# The Impact of High Glucose or Insulin Exposure on S100B Protein Levels, Oxidative and Nitrosative Stress and DNA Damage in Neuron-Like Cells

**DOI:** 10.3390/ijms22115526

**Published:** 2021-05-24

**Authors:** Adriana Kubis-Kubiak, Benita Wiatrak, Agnieszka Piwowar

**Affiliations:** 1Department of Toxicology, Faculty of Pharmacy, Wroclaw Medical University, Borowska 211, 50-556 Wroclaw, Poland; agnieszka.piwowar@umed.wroc.pl; 2Department of Pharmacology, Faculty of Medicine, Wroclaw Medical University, Mikulicza-Radeckiego 2, 50-345 Wroclaw, Poland; benita.wiatrak@umed.wroc.pl

**Keywords:** Alzheimer’s disease, hyperglycemia, hyperinsulinemia, S100B protein, oxidative stress

## Abstract

Alzheimer’s disease (AD) is attracting considerable interest due to its increasing number of cases as a consequence of the aging of the global population. The mainstream concept of AD neuropathology based on pathological changes of amyloid *β* metabolism and the formation of neurofibrillary tangles is under criticism due to the failure of A*β*-targeting drug trials. Recent findings have shown that AD is a highly complex disease involving a broad range of clinical manifestations as well as cellular and biochemical disturbances. The past decade has seen a renewed importance of metabolic disturbances in disease-relevant early pathology with challenging areas in establishing the role of local micro-fluctuations in glucose concentrations and the impact of insulin on neuronal function. The role of the S100 protein family in this interplay remains unclear and is the aim of this research. Intracellularly the S100B protein has a protective effect on neurons against the toxic effects of glutamate and stimulates neurites outgrowth and neuronal survival. At high concentrations, it can induce apoptosis. The aim of our study was to extend current knowledge of the possible impact of hyper-glycemia and -insulinemia directly on neuronal S100B secretion and comparison to oxidative stress markers such as ROS, NO and DBSs levels. In this paper, we have shown that S100B secretion decreases in neurons cultured in a high-glucose or high-insulin medium, while levels in cell lysates are increased with statistical significance. Our findings demonstrate the strong toxic impact of energetic disturbances on neuronal metabolism and the potential neuroprotective role of S100B protein.

## 1. Introduction

Alzheimer’s disease (AD), first described in 1907, is a neurodegenerative disorder characterized essentially by *β*-amyloid plaques and tangles of hyperphosphorylated tau proteins alongside cholinergic dysfunction [1,2,3]. The pathological heterogeneity characteristic of AD creates difficulties in establishing a single theory as to its cause, characterizations, and possible treatments [4]. Important risk factors include age-related biochemical and metabolic changes, vascular disease, traumatic brain injury, epigenetic factors, diet, mitochondrial malfunction, metal exposure, infections, hypertension, obesity, dyslipidemia and diabetes mellitus type 2 (T2DM) [5,6]. In diabetes mellitus, the feedback loops between insulin action and insulin secretion do not function properly. In T2DM, the pancreatic islet *β*-cell’s dysfunction, together with insulin resistance in insulin-sensitive tissues, leads to increased glucose production in the liver and decreased glucose uptake in muscle, which results in an excessive amount of glucose circulating in the blood [7]. The growing occurrence of both diabetes and dementia leading to AD is becoming a social and economic challenge worldwide [8,9].

Current research studies point out that AD and T2DM might share some underlying mechanisms and putative biochemical pathways, i.e., the desensitization of insulin signaling, most likely driven by chronic inflammation, and mitochondrial dysfunction, as a consequence of increased oxidative stress or vasculopathy [10,11]. Moreover, T2DM is a known risk factor of AD since hyperinsulinemia and insulin resistance, hallmarks of this metabolic disturbance, can lead to memory impairment [12,13]. Almost 70% of T2DM cases demonstrate diverse forms of nervous system impairment such as diabetic neuropathy, slowed digestion of food in the stomach, carpal tunnel syndrome, erectile dysfunction and other peripheral nerve problems or even central nervous system complications, including strokes and possibly cognitive impairment [14,15]. Given that numerous data points to coincidence as well as co-morbidity between those two pathological states, in 2008 prof. De la Monte [16] proposed that AD might be called “type 3 diabetes mellitus,” but this term is under criticism by others [17,18]. Nowadays, despite the vast number of papers published on AD and diabetes, the underlying link between these two disorders is still unclear [19,20]. Observations from clinical trials are highly ambiguous and have been futile in deciphering clear paths responsible for the augmented risk of dementia in T2DM patients [21,22,23]. The origin of brain neurodegeneration connected to continuous hyperglycemia is heterogeneous and comprised of modifications in neurotransmitters’ metabolism, neuronal inflammation, mitochondrial disturbances and micro- and macro-vascular dysfunction [24]. In T2DM, chronic hyperglycemia, among others, is associated with an increased risk of developing memory deficits or decreased psychomotor speed [25]. The molecular and cellular pathophysiology underlying this complication is not yet well understood. Therefore, it is important to conduct research aimed at explaining the complexity of this relationship.

S100B is an extracellular alarmin, a small helix-loop-helix protein that binds up to four Ca^2+^ per dimer in EF-hands motifs, counteracts amyloid-*β* accumulation, and is upregulated in AD [26]. Despite its most common localization in a subtype of mature astrocytes that ensheath blood vessels and in neural/glial antigen 2-expressing cells, S100B protein is also expressed in different cell types, ranging from arterial smooth muscle to melanoma cells [27]. Uniquely for the S100 protein family, S100B location is also observed in distinct subpopulations of neurons and in adipocytes, suggesting its potential new role in the regulation of energetic metabolism [28]. The levels of S100B mRNA expressed in the cerebral cortex and adipose tissue have been shown to be very similar [29]. S100B levels are raised in the adult organism as a consequence of nervous system damage, which makes it a potential clinical marker. Moreover, it was found that before any detectable changes in intracerebral pressure, neuroimaging, and neurological examination, S100B concentrations are elevated in serum or cerebrospinal fluids, enabling fast and crucial medical treatment before permanent damage occurs [30,31]. Depending on the concentration, this Janus-faced molecule can represent both beneficial and toxic effects on neuropathological changes in cellular metabolism characteristic for AD [32]. The extracellular action of the S100B protein is primarily based on its interaction with RAGE receptors, which are highly activated during T2DM or in hyperglycemic states. S100B binds to RAGE in the extracellular space and activates a number of intracellular biochemical pathways in microglia and neurons. RAGE expression is potentiated by increased extracellular concentrations of ligands, including the S100B protein. These receptors mediate both the trophic and the toxic effects of the S100B protein on cells [33,34]. At lower concentrations, the binding of S100B protein to RAGE causes activation of the Ras/ERK pathway and stimulation of MEK/MAP kinase regulated by extracellular signals, ERK/NF-κB/Bcl-2 and Ras/Cdc42-Rac1. At higher concentrations, S100B protein leads to hyperactivation of the Ras/MEK/ERK pathway, leading to overproduction of ROS [27]. Wartchow et al. [35] exposed the existence of insulin-S100B regulation of glucose utilization in the brain tissue. Moreover, it regulates glial fibrillary acidic protein and two glycolytic enzymes: fructose-1,6-bisphosphate aldolase and phosphoglucomutase in the brain, which is why it is so crucial for the potential role of S100B protein to be revealed [36,37].

The purpose of this article is to elucidate the potential role of S100B protein in correlation with oxidative and nitrosative stress in disturbed glucose/insulin homeostasis in neuron-like cells differentiated with nerve growth factor (NGF) pheochromocytoma (PC12) cells. This cellular model is widely used in both neurobiological and neurotoxicological studies to study the mechanisms of action of neurotoxicants as well as the potential for chemicals to alter neuronal differentiation [38,39]. To our knowledge, there is no information available regarding the role of S100B protein in the pathological mechanism underlying the “dance macabre” between T2DM and AD. We hypothesized that S100B levels are increased in neurons with simultaneous conditions of metabolic disturbance and neuropathological changes, and this fact could probably be used for the detection of the first cellular disruptions connected to dementia and T2DM.

## 2. Results

The MTT assay evaluated mitochondrial activity, which can be treated as a measurement of metabolic activity, and thus viability, of cells. Cell dysfunction and cytotoxicity after exposure to high concentrations of glucose or insulin were assessed by MTT assay and presented in Figure 1.

Analysis of neuron-like cells’ viability after 24 h incubation with glucose (5–500 mM) and insulin (10–750 µM) will aid the selection of optimal concentrations (40–60% viability) for further studies. The metabolic activity of neuron-like cells was decreased when they were cultured for 24 h with solutions of both glucose and insulin as calculated with reference to untreated neuron-like cells. DMSO treatment resulted in a substantial 95.6% decrease in mitochondrial activity, and a comparable reduction (89.8%) was obtained after incubation with 500 mM glucose. Administration of 10 mM, 20 mM glucose and 0.01 µM insulin did not affect the viability of neuron-like cells, as the results are similar (90.5%, 97.5%, 99.7% and 95.9%, respectively) to those obtained with untreated cells (100%). The results obtained after incubation with glucose or insulin were clearly dependent on concentration. The optimal concentrations—50–150 mM for glucose and 50–250 µM for insulin—were chosen for planned experiments based on the obtained MTT results for tested substances.

### 2.1. Nitrite Levels

The levels of cellular nitrogen free radicals were measured after 1 h and 24 h incubation with glucose or insulin. The obtained results are presented in Figure 2.

After 1 h incubation, a statistically significant increase in the level of nitric oxide was observed both in the presence of glucose and insulin. The results obtained after 1 h incubation with glucose were clearly dependent on concentration, with a statistically substantial rise after 100 mM (x-fold—1.21) and 150 mM (x-fold—1.3) glucose. In the case of insulin, all three concentrations instigated a statistically significant upsurge in nitric oxide levels after 1 h treatment compared to untreated cells. It is interesting that the addition of 250 µM insulin for 1 h produced a similar amount (x-fold—1.44) of nitric oxide to H_2_O_2_ (x-fold—1.49).

Significantly, the impact of glucose and insulin on nitric oxide levels after 24 h incubation was less expressed; only after treatment with 100 µM and 250 µM insulin were the levels of nitric oxide considerably higher (x-fold—1.13 and 1.14, respectively) in comparison to untreated cells.

### 2.2. Reactive Oxygen Species Concentration

The DCF-DA test was used to predict extracellular ROS accumulation after 1 h or 24 h incubation with glucose (50–150 mM) and insulin (50–250 µM) (Figure 3).

After 1 h incubation, a statistically significant increase in the level of ROS was observed both in the presence of glucose and insulin. A statistically significant increase in ROS levels was observed after 1 h incubation with glucose or insulin in all analyzed concentrations, with the highest concentration obtained for 150 mM glucose (x-fold—1.2). After insulin administration, the highest level of ROS was obtained after 100 µM (x-fold—0.8). No concentration dependence was observed 1 h or 24 h after glucose or insulin administration.

It is worth noting that the impact of glucose and insulin on ROS levels was less expressed after 24 h incubation, as there was an ample decrease in the level of free oxygen radicals. Furthermore, these data were supported by the results obtained with positive assay control (H_2_O_2_), confirming the observation that reactive oxygen species are the first messengers of stress in the cells, and their levels are diminished after 24 h of treatment.

### 2.3. Double-Stranded DNA Breaks

High levels of oxygen and nitrogen free radicals can lead to DNA damage, including DBSs. The number of double-stranded DNA breaks (DSBs) was assessed in the FHA by measuring the size of the nuclear halo (chromatin dispersion). The results obtained after 1 h incubation are presented on sample photos with the nuclear halo and analysis of relative NDF in Figure 4.

In neuron-like cells treated with glucose for 1 h, the increase in DBSs was generally lower than after incubation with insulin in all used concentrations. The dependence of disruption on concentration has been demonstrated for all concentrations and substances tested—the higher the concentration, the stronger the chromatin dispersion halo. The highest DBSs were observed after administration of 250 M insulin (x-fold—3.59), while the lowest DBSs levels were measured after treatment with 50 mM glucose (x-fold—2.0). The observed effect was statistically significant (*p* < 0.001) in all experimental settings.

Results obtained after 24 h incubation are presented as sample photos with the chromatin dispersion and analysis of relative NDF in Figure 5.

Similar to the results obtained in the DCF-DA, the incubation of neuron-like cells with glucose or insulin for 24 h resulted in less DNA strand damage than incubation for 1 h. After 24 h incubation with all three concentrations of insulin, the damaged DNA strand was observed to regenerate. There was a strong concentration dependence in both glucose and insulin. Glucose caused weak DNA damage with maximum levels after 24 h incubation with 150 mM glucose (x-fold—1.43). Insulin treatment had the opposite effect, leading to the regeneration of DNA double strands, with minimal values after 24 h incubation with 50 µM (x-fold—1.43).

### 2.4. S100B Protein Concentration

The study assessed the effects of glucose and insulin on the intracellular and extracellular S100B protein levels in neuron-like cells. The obtained concentrations of S100B protein after 24 h incubation with a broad range of glucose concentrations are shown in Figure 6. The extracellular concentration is presented per the amount of cells seeded in one well (1 × 10^4^), while the intracellular concentration was adjusted per 5 µg of total protein concentration in 100 µL of cell lysates measured by BCA assay.

At 50–200 mM glucose concentrations, significantly lower S100B protein levels (for around 10%) in the supernatant were observed compared to the control. In the case of extracellular S100B protein levels, there was no a dose-response relation observed. All obtained outcomes for extracellular S100B protein concentrations were statistically significant with *p* < 0.01 or *p* < 0.001. Cellular levels of S100B protein after incubation with glucose show an opposite trend compared to data obtained for supernatants, which stays in line with the results observed. As shown in Figure 6B, all treatments caused a rise in S100B intracellular concentrations, which reached their maximum after the administration of 100 mM glucose (432 pg/5 µg total protein). The increase after 50 mM glucose treatment is 23%, the next 100 mM of glucose caused a rise of 49% and, after 200 mM glucose, the upsurge of 45%. Figure 6B depicts the normal distribution of values of S100B concentrations after glucose administration confirmed by a bell-shaped dose-response dependence. Significant differences in intracellular S100B values were obtained in the case of 200 mM glucose treatment (*p* < 0.05).

Figure 7 presents the changes in S100B protein levels identified after 24 h incubation with a range of insulin concentrations. The extracellular concentration is presented per the amount of cells seeded in one well (1 × 10^4^), while the intracellular concentration was adjusted per 5 µg of total protein concentration in 100 µL of cell lysates measured by BCA assay.

The incubation with each selected insulin concentration (50–250 µM) significantly diminished the S100B protein level in the supernatants, and the strongest effect was observed after the administration of 50 µM (1769 pg/1 × 10^4^ cells). There is a weak direct proportional correlation between increasing insulin concentration and S100B protein extracellular levels with maximum concentration after 250 µM insulin treatment (1810 pg/1 × 10^4^ cells). All obtained data for extracellular S100B protein concentrations were statistically significant with *p* < 0.001. As illustrated in Figure 7B, the highest value of S100B protein was detected after administration of 100 µM insulin (341 pg/5 µg total protein), while after the administration of 250 µM insulin, the increase was the lowest compared to the control (326 pg/5 µg total protein and 219 pg/5 µg total protein, respectively). Significant differences (*p* < 0.05) in intracellular S100B values were obtained in cases of all insulin treatments. A bell-shaped graph can also be observed in the case of incubation with insulin in Figure 7B, and these values correlate favorably with data obtained with glucose. It is interesting to note that treatment incubation with insulin caused similar effects to glucose—a decrease in extracellular and increase in intracellular S100 levels.

### 2.5. Statistical Correlation between ROS, NO and DBSs Results

Correlation coefficients between DNA damage and ROS or NO levels were determined and shown in Table 1.

Based on the calculated Pearson correlation coefficients in all tested concentrations, correlations between DCF-A, FHA and Griess results for glucose after 1 h incubation were strongly positive and statistically significant, whereas correlations between the results obtained after 24 h incubation with all tested glucose concentrations were negative. After insulin treatment, a strong correlation was observed only between ROS and DNA damage after 1 h incubation as well as after 24 h treatment. In the case of insulin, there was a positive correlation only between ROS level and DNA damage after 1 h and 24 h incubation.

### 2.6. Statistical Correlation between ROS, NO, DBSs and S100B Results

Correlations between extracellular and cellular S100B protein levels and DNA damage, ROS or NO levels, analyzed by using Pearson correlation coefficients, are shown in Table 2.

Based on the calculated Pearson correlation coefficients in almost all glucose concentration tested after 24 h incubation, correlations between the results of intracellular and extracellular levels of S100B protein were negative compared to outcomes from ROS, NO and DBSs levels. The obtained correlation was strong only in the case of extracellular S100B levels vs. ROS levels, whereas after insulin administration, a strong correlation was observed only between intracellular S100B protein levels and DBSs or NO as well as among extracellular S100B protein levels and DBSs or NO. To reiterate, the correlation between extracellular and intracellular S100B levels and DNA damage or NO levels after insulin administration was strong. Such a consequent effect was not observed in a hyperglycemic environment.

## 3. Discussion

Regardless of the fact that “type 3 diabetes mellitus” has received much attention in the past decade, it is still unclear whether an overall failure of brain glucose and insulin metabolism regulations is associated with AD pathogenesis. It is generally accepted that altered cerebral glucose uptake and insulin resistance are some of the hallmarks of the progression of neurodegenerative processes [40]. Investigators from Thiambisetty laboratory were one of the first to find, in 2017, that lower rates of glycolysis and higher brain glucose levels were correlated to more severe amyloid plaques and neurofibrillary tangles found in the brains of people with AD [41]. Current evidence from the AD-T2DM animal model, where scientists combined the APP/PS1 mouse model with the genetic db/db model of type 2 diabetes, highlights neuroinflammatory processes manifested by a significant increase of microglia burden in the cortex and general brain atrophy in animals of 14 weeks of age when T2DM was started, but no AD pathology was observed [42]. Moreover, recent studies have found that AD-T2DM mice are characterized by the upregulation of a broad profile of pro-inflammatory cytokines, such as IL-1*α*, IFN-*γ* and IL-3 in the brain [43].

In our study, we demonstrated that environments with high levels of glucose or insulin have a negative impact on neuronal metabolism by elevating levels of oxidative stress markers. Our present data support the view that hyperglycemia and/or insulinemia causes activation of ROS and NO levels in neurons, which also affects double-stranded DNA breaks. Furthermore, there was a significant positive correlation between those parameters after 1 h from glucose administration. The statistical analysis did not confirm any significant differences between oxidative stress parameters DBSs after insulin treatment with two exceptions for ROS vs. DBSs after 1 and 24 h of incubation. It is suggested that, over the years, these small changes in the neuronal tissue may lead to the deactivation or shift of crucial cell cycle pathways such as the MEK-ERK1/2-NF-κB pathway as well as to the upregulation of pro-apoptotic factors, which leads to amyloid *β* accumulation. Although caution must be exerted in extrapolating in vitro findings to the in vivo situation, our data are in line with the suggestion that the fast-shifting concentrations of glucose and/or insulin in neuronal tissue may be one of the first pathological changes connected to AD and begin several years prior to the onset of clinical symptoms. Some clinical studies have suggested a significant role for S100B in neurodegeneration processes, pointing to its increased levels in the body fluids of patients with AD [44,45,46]. Christl et al. [47] estimated that S100B protein levels in cerebrospinal fluid might have a diagnostic value, particularly at the early stages of the disease, as it declines to normal levels in more advanced stages. Animal studies provide additional data showing the importance of S100B in neurodegenerative processes. The performance of S100B-overexpressing Tg2576 mice was inferior in the spatial learning study, dependent on the functioning of the hippocampus. Animals showed higher levels of brain parenchymal *β*-amyloid depositions and cerebral amyloid angiopathy, enhanced amy-loidogenic APP metabolism, augmented reactive astrocytosis and microgliosis and increased levels of pro-inflammatory cytokines (TNF-*α*, IL-1*β* and IL-6) as early as at 7–9 months of age. Mice whose gene encoding S100B was inactivated showed increased spatial memory and memorization under the influence of anxiety and increased long-term synaptic enhancement in the CA1 sector of the hippocampus. These results indicate that despite playing an important role in encouraging brain inflammatory responses, S100B has a role in directly promoting amyloidogenic APP processing [48].

Regarding the role of S100B protein in T2DM, Kheirouri et al. [49] measured its concentration in the blood serum of patients with metabolic syndromes characterized by intermittent fasting, central obesity, dyslipidemia and arterial hypertension. Moreover, the participants of the study had elevated insulin levels, showed high values on the HOMA-IR index of insulin resistance, and the serum level of S100B protein was significantly elevated compared to healthy volunteers. The extracellular action of the S100B protein is primarily based on its interaction with RAGE—the best-known class of advanced glycation end receptors [50]. S100B binds to RAGE in the extracellular space and activates a number of intracellular biochemical pathways such as MAPKs in microglia and neurons. These receptors mediate both the trophic and the toxic effects of the S100B protein on microglia and endothelial cells. RAGE is also one of the known receptors for A*β* peptide [51,52]. In the physiological state, RAGE expression in cells is kept at a low level. Increasing the amount of RAGE ligands observed in inflammation, oxidative stress, diabetes and AD results in the induction of the expression of this receptor [53]. Literature data show a relationship between glucose concentration and S100B protein secretion. Cultures of astrocytes obtained from rat brains under conditions of metabolic stress: glucose, oxygen and serum (FBS) deprivation increased the secretion of S100B protein. However, after 12 and 24 h of exposure to metabolic stress, mRNA expression for the S100B protein decreased considerably, and a significant reduction in S100B protein secretion was observed after 48 h of incubation. Based on these observations, it is hypothesized that the S100B protein may be actively secreted in the microglia in the early stages of metabolic stress [54].

The aim of our work was to broaden current knowledge on the role of S100B protein in oxidative stress instigated by local variations in glucose or insulin concentrations in neurons. We analyzed extracellular and intracellular S100B protein levels in order to evaluate the potential role of S100 protein in metabolic disturbances occurring in neuronal cells. The single marked observation to emerge from the data comparison was that neither hyperglycemic nor insulinemic conditions provoked S100B protein efflux from the neurons, and moreover, they even decreased secretion. Interestingly, a small but still statistically significant rise in S100B concentrations was observed in cell lysates. No substantial statistical correlation between intracellular and extracellular S100B levels and analyzed markers of oxidative stress and DNA damage was found after glucose administration, excluding levels in supernatants vs. ROS, where significance was observed. In contradiction, hyperinsulinemic conditions provoked a strong correlation between S100B concentration vs. NO or double-stranded DNA break levels. Collectively, these data support the notion that cellular S100B protein could behave as a neuroprotective factor against extracellular glucose/insulin fluctuations. Our observations are consistent with the results carried out on the primary cortical astrocytes of rats by Nardin et al. [55]. After 24 h, the extracellular S100B levels were reduced by around 45% in astrocytes cultured in a high-glucose medium. A decrease in glutathione content but not glutamate uptake activity was also observed. It is important to emphasize the fact that S100 protein family members exert a dual effect (neurotrophic or neuroprotective) on neurons and astrocytes, depending on the concentration attained in the brain’s extracellular space. Our findings are consistent with the previous results presented by Alhemeyer et al. [56] who showed that neurotoxicity caused by glutamate and staurosporine can be counteracted by the S100B protein. Another study performed on PC12 cells found that the S100B protein could inhibit NGF-induced cell differentiation, but increased expression of S100B did not reverse the effects of differentiation in PC12 already differentiated with NGF cells. This may suggest an effect of the S100B protein on the inhibition of cell differentiation in the early stages of cell development [57].

In conclusion, our data support the view that there is a putative relationship between S100B and metabolic disturbances in AD-like pathology and that the S100B protein acts probably as a cytoprotective factor and may protect neurons against the toxicity of local high levels of glucose or insulin during the initial phases of the pathophysiology of AD. This study is the first step towards enhancing our understanding of the role of S100 proteins in neuronal metabolism in the specific condition of “type 3 diabetes mellitus”. Further research is necessary to clarify the mechanism of action of S100B protein as well as to explain the reasons for the significant differences in their effect on neuronal properties. As the brain’s energetic regulation is entirely insulin-dependent, it would be especially interesting to evaluate the interrelation between S100 protein family members and the glucose transporters: GLUT1 and GLUT3 in the human brain as well as their relation to the receptor for insulin (IR), and these results may represent an excellent initial step toward using S100B levels as diagnostic markers in the early stages of neuropathological disorders.

## 4. Materials and Methods

### 4.1. Materials

Glucose and human insulin solutions (Sigma-Aldrich, St. Louis, MO, USA) were freshly prepared in RPMI-1640 (low glucose—5.5 mM, Biological Industries, Cromwell, CT, USA) supplemented with 2% fetal bovine serum (FBS) and 100 µg/mL penicillin-streptavidin (both from Sigma-Aldrich, St. Louis, MO, USA). Concentration ranges from 5–500 mM for glucose and 10–750 µM for insulin were chosen for the evaluation of the cytotoxicity effect. PC12 cells—pheochromocytoma cells derived from the adrenal gland of *Rattus Norvegicus*—were purchased from ATCC (Manassas, VA, USA). The differentiation was performed using human recombinant *β* nerve growth factor, NGF-*β* (Sigma-Aldrich, St. Louis, MO, USA) dissolved in PBS to a stock concentration 1 µg/mL and stored at −20 °C for up to 2 weeks.

### 4.2. Modification of the Surface of Culture Plates

The surface of the well on plates was modified using type I collagen (Sigma-Aldrich, St. Louis, MO, USA) and dissolved in 0.1 M acetic acid to a stock concentration of 0.1% (*w/v*). Before using type I collagen, a solution was dissolved in water to a final concentration of 0.01% (*w/v*), which was added to cover the surface of the wells. The 96-well plates were left at 4 °C overnight. After removing the solution, the surface was washed 3 times for 5 min with PBS (300 µL). The plates so prepared were stored at 4 °C for up to 1 month. The plates in the biological assays were irradiated with UV for 30 min before use.

### 4.3. Cell Culture Conditions and Differentiation

The differentiation process of PC12 cells was performed according to the protocol established by Greene and Tischler with some modifications [58]. PC12 cells were grown in 25 or 75-cm^2^ culture flasks in a CO_2_-incubator (37 °C, 5% CO_2_ and 95% humidity) in RPMI-1690 medium supplemented with 10% FBS and 100 µg/mL penicillin-streptomycin. The cells were used at logarithmic growth between passage 4 and 20. The PC12 cells were dissociated with TrypLE (Gibco, Thermo Fisher Scientific, Waltham, MA, USA), seeded on type I collagen-coated 96-well plates in a concentration of 5 × 10^3^ cells per well and incubated for 24 h prior to differentiation in order to let the cells adhere to the plates’ surface. For differentiation, cells were treated with 100 ng/mL of NGF-*β* freshly dissolved in RPMI 1640 media supplemented with 2% FBS and 100 µg/mL penicillin-streptavidin. For the bioassays, cells were treated with NGF-*β* for 5 days. Medium and NGF were replenished every 48 h. The differentiation process was analyzed with a holo-tomographic microscope (3D Cell Explorer, Ecublens, Switzerland). The morphological changes of NGF-treated PC12 cells versus untreated PC12 cells are shown in Figure 8. Further, the described experiments were performed after 5 days of human NGF-*β*-induced differentiation.

### 4.4. MTT Test

To determine the cytotoxicity of glucose and insulin treatment on neuron-like cells, MTT (3-(4,5-dimethylthiazol-2-yl)-2,5-diphenyltetrazolium bromide) reduction assay was performed according to the protocol of Liu et al. [59] with some modifications. This colorimetric assay is based on the ability of succinate dehydrogenase to reduce yellow MTT tetrazolium salt into blue MTT formazan crystals in living cells. The level of conversion provides an indication of mitochondrial metabolic function. Then, 5 × 10^3^ neuron-like cells were incubated with diverse concentrations of glucose or insulin for 24 h in a CO_2_-incubator (37 °C, 5% CO_2_ and 95% humidity). Next, the supernatants were removed, and the cells were washed 3 times with 100 µL of PBS to remove phenol red residue, and 100 µL of 0.5 mg/mL MTT solution in PBS was added. The cells were incubated for 4 h at 37 °C in 5% CO_2_ and 95% humidity in the darkness. Subsequently, the MTT solution was carefully discarded, the formazan crystals were dissolved in 100 µL DMSO acidified with 1N HCl, and plates were shacked for 30 min at room temperature. Cell viability was determined by measuring the absorbance at 570 nm on the Synergie multiwell scanning spectrophotometer (BIOKOM, Janki, Poland). Cell viability for each glucose or insulin concentration was calculated as the percentage of the untreated neuron-like cells. All experiments were performed on 5 wells per concentration and repeated at least 3 times.

### 4.5. Nitrite Levels Quantitation Assay

Griess reagent assay was performed to evaluate the influence of glucose or insulin on nitric oxide (NO) levels in neuron-like cells [60]. This spectrophotometric assay is based on the formation of an azo dye by the reaction of NO_2_^−^ present in the sample with the Griess reagent. For each od experimental setting, the 1 × 10^4^ cells were seeded in a 96-well plate. After 1 h and 24 h incubation with a series of glucose or insulin concentrations, 50 µL of supernatant was transferred into new 96-well plates. The 1:1 mixture (*v/v*) reagent A (1% sulfanilamide in 5% phosphoric acid) and reagent B (0.1% N-(1-Naphthyl) ethylenediamine dihydrochloride) was added to supernatants and left for 20 min in the dark at room temperature. After incubation, the absorbance at 548 nm was measured using a Varioskan LUX microplate reader (Thermo Scientific, Waltham, MA, USA).

### 4.6. Reactive Oxygen Species Concentration Measurement

To evaluate the influence of glucose or insulin on reactive oxygen species (ROS) in neuron-like cells, we used an assay with DCF-DA [61], a fluorogenic dye that measures hydroxyl, peroxyl and other ROS activity within the cell. It is deacetylated by esterases to a non-fluorescent compound, which is later oxidized by ROS into highly fluorescent 2′, 7′–dichlorofluorescein. The rest of the supernatant left on the plates used for the Griess assay was removed, and the cell pellet was washed 3 times with PBS. Next, 25 µM of DCF-DA solution in MEM medium, without supplementation or phenol red, was added to the treated PC12 cells and left for 1 h in a CO_2_-incubator (37 °C, 5% CO_2_, 95% humidity). After incubation, the DCF-DA solution was removed, cells were washed 3 time with MEM, and fresh MEM was added. The fluorescence was immediately measured with excitation at 485 nm and emission at 535 nm using a Varioskan LUX microplate reader (Thermo Scientific). All experiments were performed with 5 wells per concentration and repeated at least 3 times.

### 4.7. DNA Double-Strand Breaks Assessment

A fast halo assay (FHA) [62] was performed to assess DSBs in the DNA of differentiated PC12 cells treated with glucose or insulin. This test enables the rapid assessment of the extent of DNA breakage caused by different types of DNA lesions. After 1 h and 24 h incubation with glucose or insulin, the supernatants were collected into tubes. The trypsinization process was performed with a TrypLE solution (Gibco, Thermo Fisher Scientific, Waltham, MA, USA) for 3–5 min in a CO_2_-incubator (37 °C, 5% CO_2_, 95% humidity). After detaching from surfaces, suspended cells were collected into tubes and centrifuged for 5 min at 1000× *g* (Eppendorf, Hamburg, Germany) to get rid of cellular debris. Next, the supernatant was removed, and the cell pellet was washed with PBS and centrifuged again under the same conditions. The cells were then re-suspended at the density of 1000 cells/µL in sterile PBS and put in bathwater (37 °C). Then, 120 µl of 1.25% agarose (low melting point) in sterile PBS was added to cells and immediately sandwiched between an agarose-coated (high melting point) slide and a coverslip. After complete gelling (cooling block for 10 min), the coverslips were removed, and the slides were placed into the lysis buffer overnight at 4 °C in the dark. The next day, the slides were transferred into a Tris-HCl buffer (pH = 13) for 30 min in the dark and then twice into a neutralization buffer for 5 min. Finally, the slides were stained using 5 µM of 4′,6-diamidino-2-phenylindole (DAPI) for 20 min and analyzed under a fluorescence microscope. The DAPI-labelled DNA was visualized using a fluorescence microscope (Leica Microsystems, Wetzlar, Germany), and the subsequent images were digitally recorded on a PC and analyzed with image-analysis software developed by one of co-authors. The slides were numerically coded before reading to reduce operator bias. The extent of strand scission was quantified by calculating the nuclear diffusion factor (NDF), which represents the ratio between the total area of the halo plus nucleus and that of the nucleus. Data are expressed as relative NDF, calculated by subtracting the NDF of control cells from that of treated cells. All experiments were performed at least 3 times.

### 4.8. Human S100B ELISA Test

The S100B protein concentration was determined in the incubation medium and in the cell lysates after 24 h of incubation with glucose or insulin and measured by adapting the enzyme-linked immunosorbent assay (ELISA) kit following the manufacturer’s protocols (human S100B ELISA Kit, Genorise, England). Briefly, the standards or samples (100 µL of lysates or supernatants) were added per well, and S100B was bound by the immobilized antibody during a 1 h incubation at room temperature. After triplicate washing with Assay Buffer (300 µL each) using an auto-washer (DiaWasher ELX50, Dialab GmbH, Austria), a detection antibody specific for human S100B (100 µL) was added to the wells and incubated for 1 h in RT. Following 3 washes with Assay Buffer (300 µL each), an HRP Conjugate (100 µL) was added for 1 h in RT. After a triplicate wash with Assay Buffer (300 µL each), a substrate solution (100 µL) was added to the wells, and color developed in proportion to the amount of S100B bounded in the initial step. The color development was stopped after 20 min by adding a stop solution. The intensity of the color was measured immediately using a microplate reader set to 450 nm with subtraction of readings at 540 nm or 570 nm to correct for optical imperfections in the plate. For each od ELISA experimental setting, the 1 × 10^4^ cells were seeded in a 96-well plate. For intracellular concentration measurements, cells after incubation with glucose or insulin were centrifuged (3 min, 1200× *g*, RT), the supernatant was collected to Eppendorf vials. Cellular pellet was lysed according to freeze-thaw protocol and standardized for cellular protein concentration by BCA assay. Briefly, protein concentration was measured in cell lysates and adjusted to 50 µg protein per 100 µL. Such prepared samples were added per well and processed for ELISA assay.

### 4.9. Statistical Analysis

All experiments were carried out 3 times in triplicate. Statistical significance was calculated compared to the control. All results are presented as mean ± SEM (standard error of the mean) relative to the control—differentiated and untreated PC12 cells. Positive assay control (DMSO) was the reference value in the MTT assay. Positive assay control (H_2_O_2_) was the reference value in the Griess and DCF-DA assays. As data have normal distribution confirmed by Shapiro–Wilk test, a parametric test was used (one-way ANOVA with Tukey post hoc tests), and the Pearson correlation was performed between Griess, DCF-DA and DBS results.

## Figures and Tables

**Figure 1 ijms-22-05526-f001:**
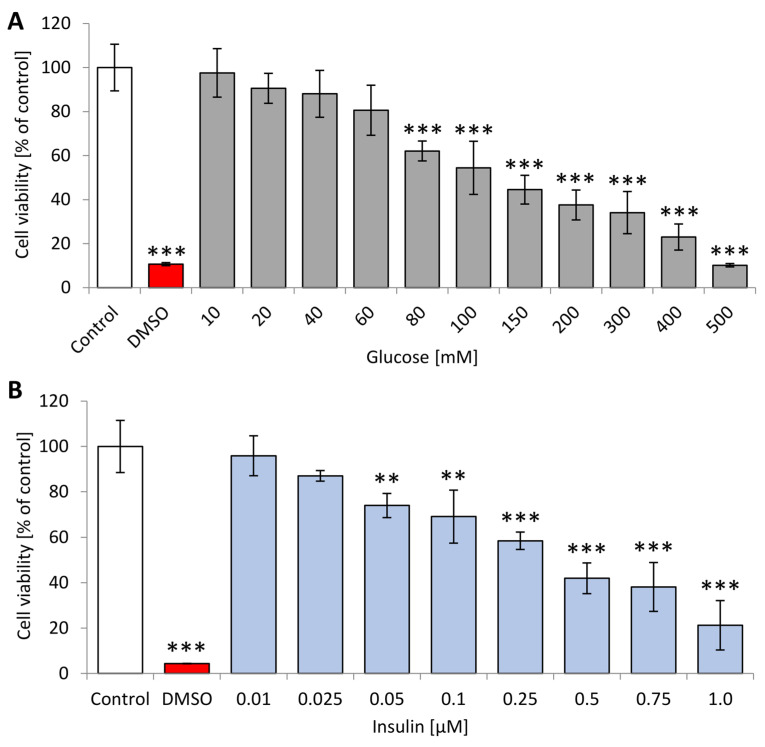
Metabolic activity measured in MTT assay of neuron-like cells after 24 h incubation with different concentrations of (**A**) glucose and (**B**) insulin. Control—untreated neuron-like cells; DMSO—PC12 cells treated with DMSO. Statistically significant differences compared to the untreated neuron-like cells: ** *p* < 0.01, *** *p* < 0.001.

**Figure 2 ijms-22-05526-f002:**
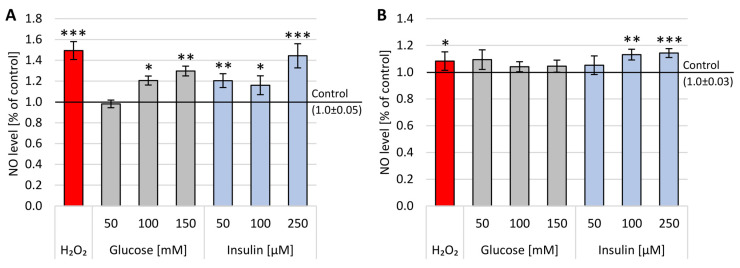
NO levels in neuron-like cells after: (**A**) 1 h incubation with glucose or insulin and (**B**) 24 h incubation with glucose (50, 100 or 250 mM) or insulin (50, 100 or 250 µM). Control—untreated neuron-like cells; H_2_O_2_—cells incubated with 50 µM H_2_O_2_ (positive control). Statistically significant differences compared to the untreated neuron-like cells: * *p* < 0.05, ** *p* < 0.01, *** *p* < 0.001.

**Figure 3 ijms-22-05526-f003:**
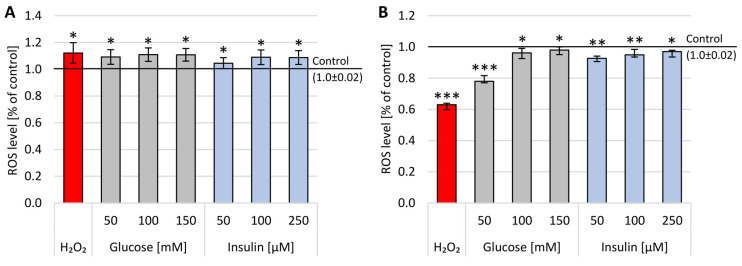
ROS levels in neuron-like cells after: (**A**) 1 h incubation with glucose or insulin and (**B**) 24 h incubation with glucose (50, 100 or 150 mM) or insulin (50, 100 or 150 µM). Control—untreated neuron-like cells; H_2_O_2_—cells incubated with 50 µM H_2_O_2_ (positive control). Statistically significant differences compared to the untreated neuron-like cells: * *p* < 0.05, ** *p* < 0.01, *** *p* < 0.001.

**Figure 4 ijms-22-05526-f004:**
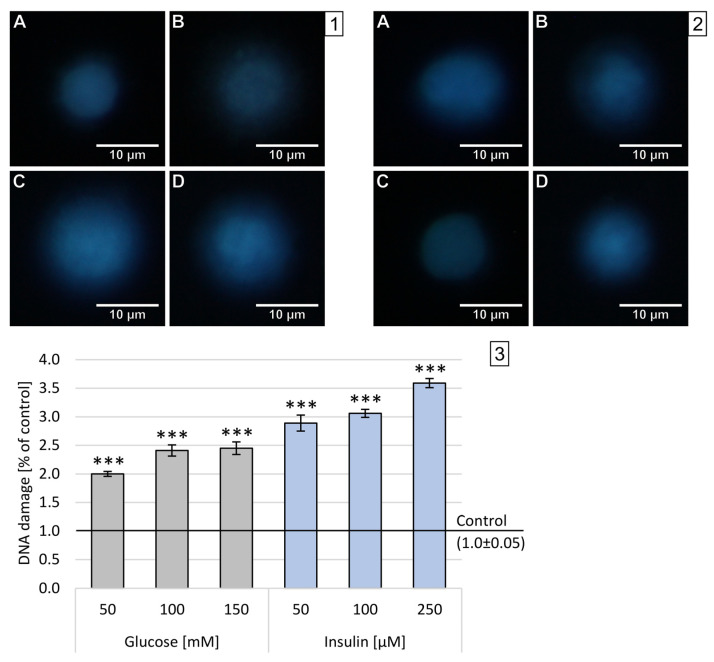
Analysis of double-stranded DNA breaks in neuron-like cells after 1 h incubation with glucose or insulin. (**1**) Sample microphotographs of a nuclear diffusion halo: A—control (untreated cells), B—glucose 50 mM, C—glucose 100 mM and D—glucose 150 mM. (**2**) Sample microphotographs of nuclear diffusion halo: A—control (untreated cells), B—insulin 50 µM, C—insulin 100 µM and D—insulin 150 µM. (**3**) A comparison of relative NDF for neuronal-like cells incubated for 1 h with glucose or insulin. Statistically significant differences compared to the untreated neuron-like cells: *** *p* < 0.001.

**Figure 5 ijms-22-05526-f005:**
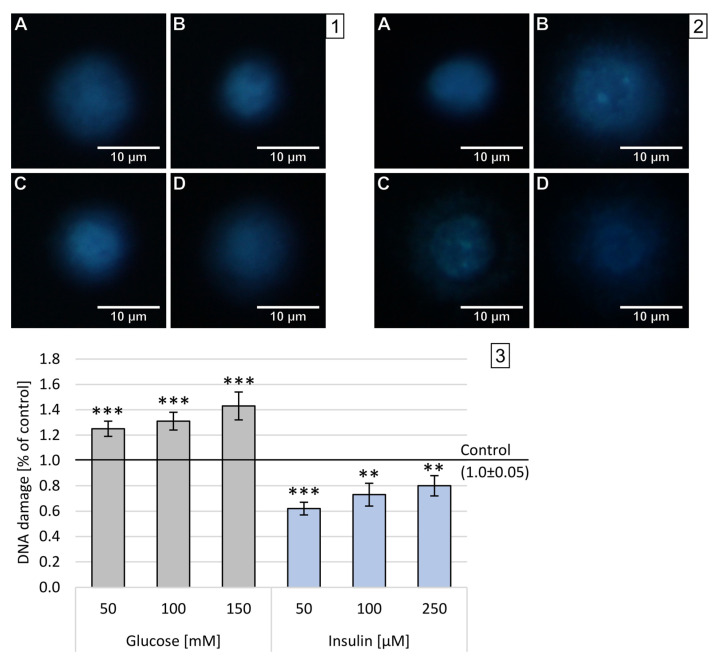
Analysis of double-stranded DNA breaks in neuron-like cells after 24 h incubation with glucose or insulin. (**1**) Sample microphotographs of a nuclear diffusion halo: A—control (untreated cells), B—glucose 50 mM, C—glucose 100 mM and D—glucose 150 mM. (**2**) Sample microphotographs of nuclear diffusion halo: A—control (untreated cells), B—insulin 50 µM, C—insulin 100 µM and D—insulin 150 µM. (**3**) A comparison of relative NDF for neuronal-like cells incubated for 24 h with glucose or insulin. Statistically significant differences compared to the untreated neuron-like cells: ** *p* < 0.01, *** *p* < 0.001.

**Figure 6 ijms-22-05526-f006:**
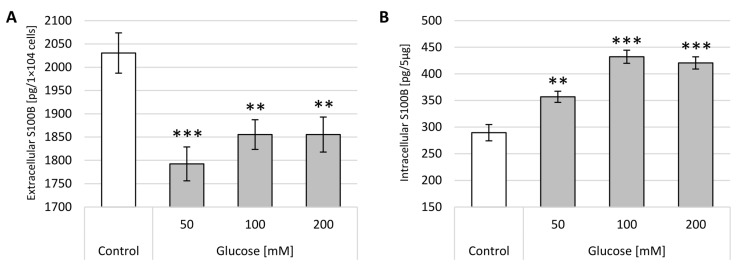
S100B protein concentration in neuron-like cells after incubation with different glucose concentrations (50, 100 and 200 mM). Control—untreated cells. (**A**) extracellular S100B levels per 1 × 10^4^ cells; (**B**) intracellular S100B levels per 5 µg of total cellular protein concentration. Statistically significant differences compared to the untreated neuron-like cells: ** *p* < 0.01, *** *p* < 0.001.

**Figure 7 ijms-22-05526-f007:**
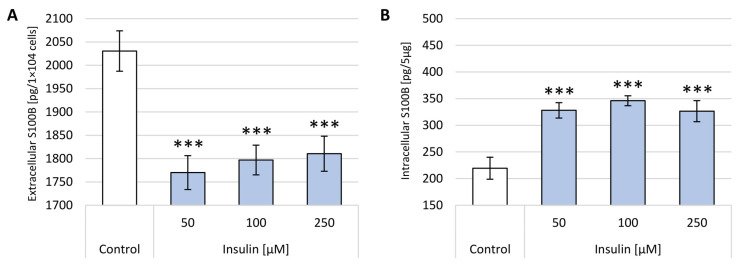
S100B protein concentration in neuron-like cells after incubation with different insulin concentrations (50, 100 and 250 µM). Control—untreated cells. (**A**) Extracellular S100B levels per 1 × 10^4^ cells; (**B**) intracellular S100B levels per 5 µg of total cellular protein concentration. Statistically significant differences compared to the untreated neuron-like cells: *** *p* < 0.001.

**Figure 8 ijms-22-05526-f008:**
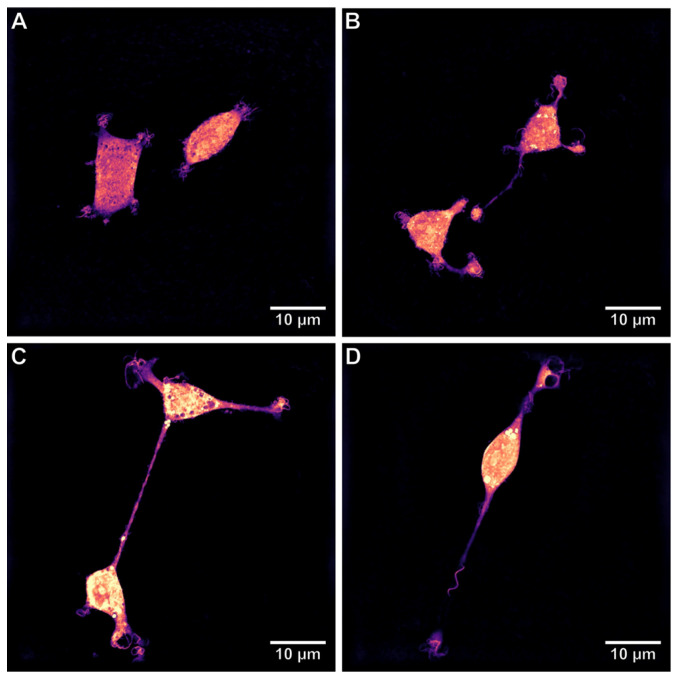
The representative microphotograph of a non-differentiated PC12 cells (**A**); the representative microphotographs of PC12 cells differentiated into neuron-like cells: after 2 days (**B**), after 4 days (**C**), and after 5 days (**D**) of incubation with human NGF-*β*. The morphological changes during the differentiation process were analyzed with the 3D Cell Explorer free-label microscope.

**Table 1 ijms-22-05526-t001:** Correlations between measured parameters ROS, NO and DBSs in neuron-like cells after glucose or insulin treatment for 1 and 24 h.

	1 h			24 h		
	ROS vs. DBSs	NO vs. DBSs	ROS vs. NO	ROS vs. DBSs	NO vs. DBSs	ROS vs. NO
Glucose	0.98	0.99	0.94	−0.70	0.44	−0.95
Insulin	0.93	0.66	0.34	0.96	−0.65	−0.42

**Table 2 ijms-22-05526-t002:** Correlations between 100B protein levels and measured parameters: ROS, NO and DBSs in neuron-like cells after glucose or insulin treatment for 24 h.

	Intracellular S100B			Extracellular S100B		
	vs. DBSs	vs. NO	vs. ROS	vs. DBSs	vs. NO	vs. ROS
Glucose	−0.62	−0.62	0.33	−0.99	−0.99	0.92
Insulin	0.90	0.90	−0.77	0.98	0.98	−0.60

## Data Availability

The data generated and analyzed during the current study are available from the corresponding author upon reasonable request.

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
