# Peer review of "The Impact of High Glucose or Insulin Exposure on S100B Protein Levels, Oxidative and Nitrosative Stress and DNA Damage in Neuron-Like Cells"

_ijms, 2021, doi:10.3390/ijms22115526_

Round 1

Reviewer 1 Report

This paper reports on the effect of hyper-glycemic and -insulinemic conditions on the level of S100 protein. This topic is interesting and so far has been studied very scarcely. Since there is a positive correlation between long-term diabetes and dementia, the interplay between diabetes and incidence of AD is of great importance to study. 

Comments: 

1. Authors state that the only strong correlation has been observed between ROS and DNA damage following both insulin and glucose treatment. What are the radical species responsible for such a correlation, if no correlation has been found between nitric oxide and DNA damage (after 1 hour). 

2. Is there any observable ROS-induced S100 damage? 

English throughout the MS should be carefully checked. e.g. "This results...", "Our observations are..." and many more. 

Author Response

Reviewer 1:

  1. Authors state that the only strong correlation has been observed between ROS and DNA damage following both insulin and glucose treatment. What are the radical species responsible for such a correlation, if no correlation has been found between nitric oxide and DNA damage (after 1 hour). 

Ad. 1 DNA damage may be related to the increase in oxygen or nitrogen free radicals. In our study, we showed a strong correlation between DNA double strand breaks and ROS/NO after short-term exposure to high glucose levels but not after 24h incubation period. These results highlighted that DNA damage can be oxidative stress related in short-time glucose treatment. When the high glucose environment is left for a longer time, the stand breaks are not ROS or NO related. In case of hyperinsulinemic conditions, weak correlation between NO and DBS after 1h and 24h treatment with insulin shows that only radical species specifically measured in DCF-DA test which are hydroxyl and peroxyl radicals could be responsible for DBSs damage in this experimental setting. Remarkably, we have observed a strong correlation between DBSs and ROS in both, short and long term exposure which opens a new experimental path to see which radicals are generated in neuronal cells in case of hyper-glycaemic and –insulinemic conditions.

  1. Is there any observable ROS-induced S100 damage? 

Ad. 2 We haven’t check it as are main goal was to see how hyper- glycaemic and -insulinemic states influence S100B levels. Treatment of C2C12 myoblasts for 72 h with 100µM Paraquat, a herbicide known to cause ROS overproduction lead to increased S100B levels but I haven’t found papers on neuronal like cells concerning this subject (Morozzi, G. et al. 2017. Cell Death Differ 24, 2077–2088). Leclerc et al. (2007, J. Biol. Chem. 282/43, 31317–31331) has measured the time-dependent formation of ROS in the cells in the presence of 5μM S100B. Similar to theirs results, in our work the increase in ROS level was only significant 1h after exposure to S100B. It has been shown that ROS is the critical factor in S100A8/A9-induced cell death (Lim SY. et al. 2009. J Leukoc Biol. 86(3):577-87.). Increased NADPH activity and ROS production were observed in activated cell lines overexpressing S100A8/A9 (Benedyk, M. et al. 2007. J. Invest. Dermatol. 127, 2001-2011).This is a very interesting scientific direction which we will plan to incorporate in ours future experiments.

English throughout the MS should be carefully checked. e.g. "This results...", "Our observations are..." and many more. 

The whole paper was send to the external translator for language correction.

Reviewer 2 Report

In this MS, Authors Adriana Kubis-Kubiak and collaborators, investigated the impact of hyper/glycemia and insulinemia on oxidative stress, nitrite oxide, S100B protein levels and DNA damages in neuron-like cultrured cells.

This is an interesting MS, however I have several major points:

  • Authors should consider revise the title of the MS and remove abbreviations.
  • Abstract line 9. The sentence starting with “The mainstream concept of AD neuro-

pathology based on…” is very long and should be simplified.

  • It is not usual to see references in an abstract. Here, there are three references.
  • Lane 17 “we established” is not a correct description of the scientific research. Passive tense is usually chosen with “measured” or “quantified” terms.
  • An abstract should be self-explanatory enough for the reader. Aim of the study is unclear and rational to evaluate S100B levels absent.
  • Introduction section should be reduced.
  • Figure 1, I think it is better to show increasing concentrations on x axis from the left to the right.
  • In Fig 1A, what medium was used for the control condition? As regular medium contains 5.5 mM glucose how Authors can perform their control and 5 mM conditions?
  • Line 131 page 4 Authors wrote “an optimal concentrations (40-80% viability) for further studies.” I am very surprised to see such big range to define “optimal concentrations”. In addition, do conditions that lead to more than 50% cell mortality be considered as an optimal, relevant concentration?
  • In figure 3A, I understand positive control is H2O2 treated cells. Enhanced ROS in H2O2 treated cells is not evident at all.
  • In figure 3B why ROS level is very low in H2O2 treated cells?
  • Images on figure 4 are lacking a scale.
  • On figure 6 and 7 results should be expressed in pg per mg cellular protein. Expression of protein quantity per mL is not correct as it does not take into account cell number. Please note the treatment induced 40 to 80 % cell mortality.

Author Response

Reviewer 2:

  1. Authors should consider revise the title of the MS and remove abbreviations.

Ad. 1

The title was changed according to reviewer’s comment New title - ”The impact of high glucose or insulin exposure on S100B protein levels, oxidative stress and DNA damage in neuron-like cells”

  1. Abstract line 9. The sentence starting with “The mainstream concept of AD neuropathology based on…” is very long and should be simplified.

Ad. 2 The sentence was divided into two “The mainstream concept of AD neuropathology based on pathological changes of amyloid β metabolism and formation of neurofibrillary tangles is under criticism as Aβ-targeting drug trials failed. Recent findings has shown that it’s highly complex disease involving a broad range of clinical manifestations as well as cellular, and biochemical disturbances.”

  1. It is not usual to see references in an abstract. Here, there are three references.

Ad. 3 The references where removed from abstract.

  1. Lane 17 “we established” is not a correct description of the scientific research. Passive tense is usually chosen with “measured” or “quantified” terms.

Ad. 4 The term established was replaced with measured.

  1. An abstract should be self-explanatory enough for the reader. Aim of the study is unclear and rational to evaluate S100B levels absent.

Ad. 5 The abstract was edited and an appropriate explanation of the purpose of the study was added to the abstract.

  1. Introduction section should be reduced.

Ad. 6 The introduction paragraph was shortened from 1140 to 952 words.

  1. Figure 1, I think it is better to show increasing concentrations on x axis from the left to the right.

Ad. 7 Figure 1A and B were changed according to reviewer comment to show increasing concentrations of glucose or insulin on X axis.

  1. In Fig 1A, what medium was used for the control condition? As regular medium contains 5.5 mM glucose how Authors can perform their control and 5 mM conditions?

Ad. 8 This is a very good comment. As control (normoglycaemic state) we have used cells in growth media  with 5.5mM glucose. Neuronal cell lines are very sensitive to glucose levels in media so it would be difficult to plan experiments with no glucose in environment. Concluding, we have rearranged the Figure 1A. Thank you very much for this very proper observation.

  1. Line 131 page 4 Authors wrote “an optimal concentrations (40-80% viability) for further studies.” I am very surprised to see such big range to define “optimal concentrations”. In addition, do conditions that lead to more than 50% cell mortality be considered as an optimal, relevant concentration?

Ad. 9 There is an spelling error. There should be written between 40-60% of viability as for the next experiments there were chosen the concentrations: 100mM (54,4% ) and 150mM (44,5%) for glucose; 100µM (69%) and 250µM for insulin (58,5%). As we didn’t know what to expect with result of S100B protein (it can work as pro and anti-inflammatory factor) and wanted to have proportional increase in concentrations, we have decided to choose also 50mM (67%) glucose and 50µM insulin (73,9%) which causes weak cytotoxic effects.

  1. In figure 3B why ROS level is very low in H2O2 treated cells?

Ad. 10 It is because NO levels rise very quickly as you can see it after 1h incubation where the levels are much higher after H2O2 treatment comparing to those obtained after 24h. This is due to the fact that nitric oxide radicals are very unstable and as we were performing our experiments in cell culture medium, it’s possible that during this long period they have reacted with the different salts which are included in the medium.

  1. In figure 3A, I understand positive control is H2O2 treated cells. Enhanced ROS in H2O2 treated cells is not evident at all.

Ad. 11 We have repeated the experiments (all together tenfold) and obtained values which you can see on  newly created figure 3B. We have detected and excluded outliers from the presented data by performing Grubbs test. We have shown the data for ROS levels after 24h to express the fact that the effect observed was fast and strong. As reactive oxygen species are transient mediators of oxidative stress in cells we didn’t observed it for a longer period of time and we wanted to emphasize this observation by showing a data obtained after 24h incubation period.

  1. Images on figure 4 are lacking a scale.

Ad. 12 The scale bar was added on the figure 4 and 5 as well as the information about the units was added to the legend.

  1. On figure 6 and 7 results should be expressed in pg per mg cellular protein. Expression of protein quantity per mL is not correct as it does not take into account cell number. Please note the treatment induced 40 to 80 % cell mortality.

Ad. 13 Thank you very much for this comment. Yes, you are right. We should calculate protein concentrations with e.g. BCA assay and then calculate it pg S100B/mg of total protein. As we explained in annotation to the point 9 the highest mortality was 55,5% for 250mM insulin and the lowest 26% for 50µM insulin. We will surely take your comment under account in our future experimental procedures.

Round 2

Reviewer 2 Report

This MS is a revised version from Adriana Kubis-Kubiak and collaborators who investigated the impact of hyper/glycemia and insulinemia on oxidative stress, nitrite oxide, S100B protein levels and DNA damages in neuron-like cultured cells.

In their revised version, Authors addressed most of my points. Still few points remains to be considered:

  • Page 4. I am concerned by “the selection of optimal concentrations (40-60% viability)”. In all experiments, half the treated cells are dead. Furthermore, authors did not express the results per quantity of cellular protein, nor living cells. In such conditions how can Authors propose a specific mechanism for their treatment as half of the cells are dead?
  • Figure 3, Authors wrote “incubation with  20µl  of  H2O2  (positive  assay  control)”. First, I am not sure that writing a volume constitutes a scientific way to express a treatment condition. Second, I am not convinced by the answer concerning positive control in figure 3 (H2O2 treated cells). If increase in ROS (3A) is very weak but apparently significant why authors use same condition to define their “positive control” in figure 3B. In 3B, the positive control gives negative results. A positive control is a condition where positive results are expected…

  • I understand after results displayed on figure 1, Authors chose concentrations leading to about 50% mortality (50-150 mM for glucose and 50-250 µM for insulin. Why very high concentrations ranges are displayed in figure 6 and 7?

  • The MS still contains typos. For exemple page 13 “that S100B protein, acts probably acts as a cytoprotective factor”

  • Figure 8 is lacking scale.

Author Response

Dear Reviewer 2,

Underneath, you can find our explanations to points highlighted. We hope that this time you will be satisfied with our improvements.

Best regards

Adriana Kubis-Kubiak

Page 4. I am concerned by “the selection of optimal concentrations (40-60% viability)”. In all experiments, half the treated cells are dead. Furthermore, authors did not express the results per quantity of cellular protein, nor living cells. In such conditions how can Authors propose a specific mechanism for their treatment as half of the cells are dead?

Ad. Page 4 The gold standard in cytotoxicity studies and basic information needed before starting a crucial part of the experiment is to establish LC50, which is the concentration in which 50% of the cell after treatment is dead/alive. Only after establishing such conditions, we can check how the potential cytotoxic substance is influencing the specific metabolic pathways of cells. Our study aimed to create hyper-glycemic or -insulinemic conditions that will influence cell metabolism and viability. If we have used the concentration that does not cause the mortality it would NOT be hyper -glycemic and -insulinemic condition. To broaden the range of the experimental settings we have used concentrations around LC50 which are 40-60% viability, which is generally used and accepted method when the output of the final experiments is unknown. As S100B protein can be secreted by living and dying cells we wanted to compare untreated (1000% live) cell’s 100B secretion with secretion obtained from cells under glycemic and insulinemic stress.

Regarding the comment about presenting the results per quantity of cellular protein. The 100B concentrations were measured in cell supernatant and lysates. As the experiment was performed in medium with 2% FBS the bovine serum albumin and globulins will be the most abundant protein in the supernatants. The S100B concentration constitutes maybe of 0,1% of all proteins in cellular supernatants. Furthermore, I haven’t found literature showing levels of different proteins measured by ELISA assay in cell supernatants calculated per protein concentration. It is presented per ml of fluid [examples 1-4]. As we understand that it doesn’t speak to you so we have changed it for cellular number ([5] likewise), which was in all wells the same - 1x104, always keeping in mind the fact that S100B protein is secreted from alive as well as dying cells.

  1. Fig. 1 in Sugihara, S. et al. (2020). Increased diagnosis of autoimmune childhood-onset Japanese type 1 diabetes using a new glutamic acid decarboxylase antibody enzyme-linked immunosorbent assay kit, compared with a previously used glutamic acid decarboxylase antibody radioimmunoassay kit. J Diabetes Investig, 11(3), 594–602.
  2. Fig. 4A, 4E in He, Lf., Wang, Tt., Gao, Qy. et al. (2011) Stanniocalcin-1 promotes tumor angiogenesis through up-regulation of VEGF in gastric cancer cells. J Biomed Sci 18, 39
  3. Fig. 4A/B in Sarkar S et Al. (2020) Comparison of VEGF-A secretion from tumor cells under cellular stresses in conventional monolayer culture and microfluidic three-dimensional spheroid models. PLoS One. 2020;15(11):e0240833.
  4. Fig. 5 in Schmidt NM. Et al. (2021) Targeting human Acyl-CoA:cholesterol acyltransferase as a dual viral and T cell metabolic checkpoint. Nat Commun. 12: 2814.
  5. Fig. 4C in He F. et al. (2021) ATF5 and HIF1α cooperatively activate HIF1 signaling pathway in esophageal cancer. Cell Commun Signal. 19: 53.

In the case of cell lysates, we have repeated experiments with adjusting the cellular protein concentration. We have centrifuged the cells, discarded supernatants, disrupted cellular membranes by freeze-thaw protocol, and measured protein concentrations in the samples. We have adjusted the concentration with PBS to have 5µg in 100µl and such prepared samples were taken on ELISA assays. The obtained results are shown in Fig 6B and 7B.

Figure 3, Authors wrote “incubation with  20µl  of  H2O2  (positive  assay  control)”. First, I am not sure that writing a volume constitutes a scientific way to express a treatment condition. Second, I am not convinced by the answer concerning positive control in figure 3 (H2O2 treated cells). If the increase in ROS (3A) is very weak but apparently significant why authors use the same condition to define their “positive control” in figure 3B. In 3B, the positive control gives negative results. A positive control is a condition where positive results are expected…

Ad. Figure 3 We agree with the Reviewer's comment that it would be appropriate to use the H2O2 concentration instead of the volume – we have corrected this in the manuscript. When it comes to what we regard as a positive and negative control, we consider it appropriate to use the approach that we presented in our study. The negative control is our benchmark against which we compare the results. In turn, in the positive control, we apply an active substance that causes a visible effect – regardless of whether the effect is harmful or beneficial. In addition, changing the reference point for individual tests, as proposed by the Reviewer, in our opinion, would introduce unnecessary confusion and make the results difficult to understand.

I understand after the results displayed in figure 1, Authors chose concentrations leading to about 50% mortality (50-150 mM for glucose and 50-250 µM for insulin. Why very high concentrations ranges are displayed in figure 6 and 7?

Ad. As already mentioned, we have chosen the concentration close to LC50, but as these experiments were performed concomitantly, we also have data for higher concentrations. We have edited graphs 6 and 7 by displaying only the concentrations used for other experiments. We have added axis lines.

The MS still contains typos. For example page 13 “that S100B protein, acts probably acts as a cytoprotective factor”

The whole article was checked for grammar, spelling, and punctuation errors with an online grammar checker which has found only 4 spelling mistakes and 8 punctuation errors.

 Figure 8 is lacking scale.

The scale was added to Figure 8

Round 3

Reviewer 2 Report

In their revised version of the MS, Authors have addressed my points.